# Exploring the Limitations of Graph-based Logical Reasoning in Large Language Models

## Abstract

Pretrained Large Language Models have demonstrated various types of reasoning capabilities through language-based prompts alone. However, in this paper, we test the depth of logical reasoning for 5 different LLMs (GPT-4, GPT-3.5, Claude-2, Llama-2 and Palm-2) through the problems of graph reasoning. In particular, we design 10 distinct problems of graph traversal, each representing increasing levels of complexities. Further, we analyse the performance of models across various settings such as varying size of graphs as well as different forms of k-shot prompting. These models are evaluated using two distinct metrics – absolute accuracy, which evaluates model responses using a binary label (Correct/Wrong), and partial credit, which evaluates the sequence of predicted nodes against the actual solution step by step, and awards score for the number of nodes correctly predicted before deviating from the correct response. We find that apart from certain powerful, language models do not possess strong reasoning capabilities The reasoning capabilities has an inverse relation to the average degrees of freedom of traversal per node in graphs. Further, we note that k-shot prompts has an overall negative impact on the reasoning abilities of language models. We finally conclude that powerful models (including GPT-4, Claude-2 and GPT-3.5) possess an estimated variable tracking depth of less than 10 nodes, making them unsuitable for complex reasoning tasks.

## 1 Introduction

Large language models (LLMs) powered by deep learning have made significant advances in recent years, demonstrating impressive natural language processing capabilities. These models can generate human-like text and engage in layered contextual conversation. This has led to excitement about the potential for LLMs to exhibit intelligent behavior and assist humans in a variety of tasks involving language and reasoning. However, there is debate around whether the latest iterations of LLMs actually "understand" language or display true reasoning abilities. While they can certainly generate coherent text, it is unclear whether LLMs construct valid underlying relation graphs that carry true logical reasoning.

Various studies have evaluated the different forms and aspects of reasoning in LLMs, including mathematical reasoning Yuan et al. (2023), semantic reasoning Shridhar et al. (2022), contextual deduction reasoning Bian et al. (2023), as well as general algorithmic reasoning Zelikman et al. (2023). This paper seeks to carefully evaluate the specific problem of graph-based logical reasoning, which involves many aspects of reasoning including multi-hop reasoning, memory state-tracking Adhikari et al. (2020) as well as sub-graph evaluation Choudhary & Reddy (2023). Due to its unique nature, this problem highlights many interesting properties about LLMs.Reasoning is defined here as the ability to make logical inferences, resolve ambiguities, incorporate background knowledge, and demonstrate practical judgement when responding to linguistic inputs.

Wang et al. (2023) dived into various properties of graph reasoning through graph traversal problems. Through this study, various interesting properties, such as preliminary reasoning capabilities in LLMs in basic graph traversal problems, and the unhelpful nature of advanced prompting techniques for logical deduction problems. While this is a comprehensive study on its own, we build on top of this work to study the problem through wider and more granular approach. Specifically, we carry out this evaluation through a series of increasingly complex graph problems and settings. Par-

ticularly, we construct 10 distinct graph problems requiring multi-hop reasoning and tracking. This includes tree-based graph traversals, grid-based graph-traversals as well as certain special classes of problems. We evaluate five different LLMs (GPT-3.5, GPT-4, Claude-2, Llama-2 and Palm-2). While the OpenAI family of models is thoroughly investigated in various papers, this paper studies the nature of various other contenders. For example, in this paper, we clearly demonstrate that Anthropic's Claude-2 is a good logical reasoning model, only sub-standard to GPT-4 among the various models publically available for access. We also choose adopt certain design choices that highlight biases in the training of various models. This includes methods like jumbling the sequence of nodes, as well as testing if the models can "admit" that a solution can not be found, or in other words, testing if the models are biased to responding with a non-negative response.

Overall, through this work, we highlight the following properties about LLMs

1. **Ranking various models for their reasoning abilities:** Through various experiments and inferences, we conclude that there is a clear hierarchy among the various models in terms of reasoning strength. In general, the ranking can be summarized as the follows (in decreasing order of reasoning strength) – GPT-4, Claude-2, GPT-3.5, Llama-2 and Palm-2.

2. **LLMs are shallow state memory trackers**: LLMs can only partially reason over simpler graph problems, and quickly deteriorate when the size, or averege degrees of freedom of traversal per node of the graph increases. Through this, we estimate that even the strongest reasoners (like GPT-4) only have the ability to track variables to a depth of less than 10 states. Further, we provide evidence that the parametric size of a model is directly proportional to its memory-tracking ability, subject to its training method. This, thus, limits the various LLMs available to simpler reasoning tasks.

3. **LLMs performance is inversely correlated to the degrees of freedom per node** As a result of shallow memory tracking ability, LLMs fail to track multiple possible divergences (branches) from a node. We observe that when the average number of branches per nodes increases above a certain threshold, LLMs start predicting wrong traversal paths.

4. **GPT-4, Claude-2 and GPT3.5 have some preliminary reasoning abilities, while Llama-2 and Palm-2 are, in general, " data replicators"** . While GPT-4, Claude-2 and GPT-3.5 may not be capable to fully solve complex problems, we observe that they start in the right direction before deviating from the correct path. This demonstrates that these models indeed do adopt valid fundamental reasoning pathways. On the other hand, we observe that Llama-2 and Palm-2 do not perform much better than random pathway selections (sometimes even worse), which indicates a training data "overfit".

5. **increasing k-shot examples have a negative effect on logical deduction**: In various cases, we observe that 3-shot settings perform worse than, if not at par with, 1-shot settings. Through this, we deduce that LLMs perform better with fewer k-shot examples.

## 2 EXPERIMENTAL SETTING

A typical graph traversal problem involves navigating a graph $G = \{V, E\}$ between a specified pair of nodes $u, v \; \epsilon \; V$ through a sequence of edges $e = (e_1, e_2....e_n) \; \epsilon \; E$. Traversal of a graph $G$ can be formulated in various ways, such as determining connectivity between two sub-graphs, shortest/least-cost path optimization and maximum flow optimization, and so on. However, we identify shortest/least-cost path optimization as one major class of problems involving multi-hop non-trivial reasoning. Analyzing the properties of large language models on such path optimization problems can thus reveal many interesting properties of the models, apart from multi-hop reasoning ability, such as multi-variable state tracking ability, recursive thinking as well as the ability to reject non-optimal paths. We start by defining the types of graphs that are considered for evaluating traversal properties in LLMs.

### 2.1 DEFINING GRAPH LEVELS AND COMPLEXITY

As a starting point, we categorize various types of graphs into two major categories, as described below.

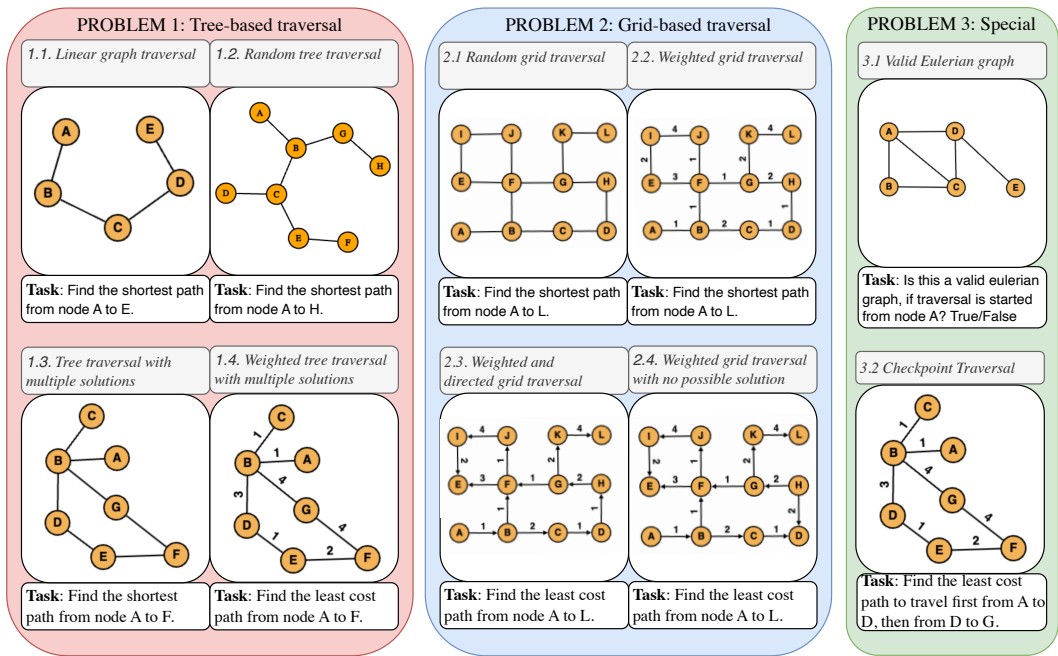

Figure 1: Visualization of all problem categories considered for evaluating LLMs. For each problem, we create 3 variations – *O(10)*, *O(20)* and *O(20) jumbled*, representing increasing levels of difficulty.

**Problem 1: Tree-based graphs:** these graphs are defined as connected undirected graphs, where any two nodes $u, v$ are connected by atmost one edge $e\epsilon\{0, R^+\}$, (where $R^+$ represents positive real number values). In the above representation $e = 0$ represents no connection, $e = 1$ represents a simple unweighted connection between nodes $u, v$, and all other positive values $e \epsilon (R^+|e \neq 1)$ represents a weighted connection between nodes $u, v$. By definitions, no cycles exist in a tree-based graph. Based on this, we formulate the following problems

- **Problem 1.1: Linear graph traversal:** a linear graph is defined as a graph $G(n)$ with $n$ nodes such that its nodes $V = (v_1, v_2, ...v_n)$ are connected pairwise sequentially ($e_{i,i+1} = 1$), where $e_{i,j}$ represents an edge between nodes $v_i$ and $v_j$. The task involves traversing from node $v_1$ to $v_n$. This is the most trivial form of graph traversal, since there exists only one possible traversal path originating from node $v_1$.

- **Problem 1.2: Random tree traversal**: given a unweighted and undirected tree-based graph $G(n)$ of order $n$, the task involves traversal from node $v_1$ to $v_n$. This involves traversing through and rejecting many dead-end paths. We ensure that there only exists one possible path from node $v_1$ to $v_n$.

- **Problem 1.3: Tree traversal with multiple possible solutions:** given a unweighted and undirected tree-based graph $G(n)$ of order $n$, the task involves finding the shortest path from node $v_1$ and $v_n$, among many possible paths from $v_1$ and $v_n$. There exists only one shortest possible path in this tree.

- **Problem 1.4: Weighted tree traversal with multiple possible solutions:** this setting is similar to problem *1.3*, except the edges are weighted. Hence, the task is modified into a least-cost traversal problem. There exists only one possible least cost path.

**Grid-based graphs:** these graphs are defined as two-dimensional connected graphs $G(M, N)$ of size $M \times N$, where node $u = G(i, j) \ \forall(i, j)|(1 \leq i \leq M), (1 \leq j \leq N)$ is connected to another node $v = G(p, q)$ by atmost one edge $e\epsilon\{0, R^+\}$, (where $R^+$ represents positive number values, and the definition of edge values are similar to tree-based graphs). A grid-graph, by virtue of being two-dimensional, is more densely connected than a tree-based graph, and inherently has cyclic connections (loops) between nodes. Thus, these form a more-challenging class of graph-

traversal problems due to the involvement of selection among a higher number of possible solutions. Based on this, we formulate the following problems.

- **Problem 2.1: Random grid traversal:** given a unweighted and undirected graph $G(M, N)$ with dimensions $(M, N)$, the problem involves finding the shortest path of traversal from node $G(1, 1)$ to node $G(m, n)$. There exists only one shortest possible path per grid.

- **Problem 2.2: Weighted grid traversal:** This problem is similar in setting to problem *2.1*, except the edges are weighted. Hence the task evolves into a least-cost path traversal from node $G(1, 1)$ to $G(m, n)$. There exists only one least-cost path per grid.

- **Problem 2.3: Directed and weighted grid traversal:** This problem is similar in setting to problem *2.1*, except all edges are weighted as well as directed. Hence the problem evolves into finding a valid least-cost traversal from node $G(1, 1)$ to $G(m, n)$.

- **Problem 2.4: Directed grid traversal with no possible solution:** This problem involves a directed grid $G(M, N)$, such that no valid path exists because of randomized directions. The goal of the problem is to evaluate whether LLMs are capable of evaluating directional conflicts, and come to the conclusion that no path is valid.

By incrementally adding constraints to both the problem categories above, we hence create an analysis framework to determine the graph reasoning abilities of LLMs in increasing levels of complexity. Even between the two problem categories, problem 2 poses a more challenging problem than problem 1, due to a larger number of average degrees of freedom of traversal per node.

**Problem 3: Special problems:**     Apart from the two aforementioned categories of problems, we define two special traversal problems in the context of tree-based problems.

- **Euler walk**: Given a valid euler graph $G$, an euler path is defined as a path which covers all edges $E$ of the graph exactly once. The eulerian path traveral is a more viable alternative to the *hamiltonian path* traversal, which involves travelling all nodes $U$ of the graph exactly once, since the former can be evaluated computationally by a finite set of conditions, unlike the latter, which is an NP complete problem, and cannot be evaluated using a polynomial time algorithm. Based on this, we formulate the problem as follows.

  - **Problem 3.1: Valid Euler graph:** Given a tree-based graph $G(n)$ with $n$ nodes, the task involves identifying whether or not a valid euler path is possible, given a starting node $v_i$.

- **Checkpoint traversal:** This problem involves traversal of a tree-based graph $G$ between two nodes $u$ and $v$, such that a specific node $w$ must be part of the traversal path. Mathematically, this is equivalent to breaking down the traversal into two distinct problems – traversal from $u$ and $w$, and traversal from node $w$ and $v$. The goal of this cascaded traversal problem is to evaluate whether a single LLM response can solve more than one problem simultaneously. We hence formulate the following problem.

  - **Problem 3.2: Cascaded graph traversal:** Given a weighted tree graph $G(n)$ with n nodes, find the least cost path to traverse from node $v_1$ to a randomized node $v_i$, and then from $v_i$ to $v_n$.

## 2.2   GRAPH GENERATION, PROMPTING AND EVALUATION

**Automated graph generation:**   Graphs, as well as corresponding solutions for all problems are automatically generated, in the form of adjacency matrices with nodes labelled alphabetically. We ensure that for all problems, there exists only a unique solution for all problems, so that evaluation can be unambiguous. For problems *3.1* (Euler walk), since there can be no guarantee of a single eulerian path, we convert the problem into a True/False problem. We also ensure that the distribution of valid and invalid euler graphs is balanced. All remaining problems require LLMs to provide a sequence of nodes in response. Thus, while lower LLM accuracies are expected, a more comprehensive evaluation can be performed. Across all problems, we generate three variations (or orders) of problems, depicting increasing levels of complexities.

1. *O(10)*: This refers to graphs that have a total number of nodes varying around $n = 10$, or between 5 and 15 (inclusive) to be exact. Hence the problem is formulated as traversal between *A* and *J* when $n = 10$, and so on, in the case of problem categories 1 and 2.

2. *O(20)*: This refers to graphs which have a total number of nodes varying around $n = 20$, or between 16 and 26 (inclusive). Hence, when $n = 20$, the problem is formulated as traversal between node *A* and *T*, and so on, in the case of problem categories 1 and 2.

3. *O(20) jumbled*: Unlike the previous two cases, where the labels of the nodes are more or less in order when traversed depth first from the starting node, we now jumble the labels of the nodes in *O(20)*, to make the problem more complex. Hence, while the rows and columns of the adjacency matrix are still alphabetically ordered, the solution node sequence is completely randomized. This helps us evaluate whether LLMs rely on training biases such as order of nodes to return solutions.

Further, we ignore evaluating problem 3.1 (valid euler graph) for O(20) and O(20) jumbled graphs. This is because generating larger graphs with guaranteed eulerian conditions are computationally highly expensive.

**Prompting techniques:** For prompting, our goal was to ensure that all models are given the same prompts for fair evaluation. The format of prompts is included in the Appendix. For all models, we test graph reasoning abilities across 3 k-shot settings – *zero-shot* setting ($k = 0$), *one-shot* setting ($k = 1$) and *three-shot* setting ($k = 3$) Brown et al. (2020). Since our goal is to evaluate the raw graph reasoning abilities of LLMs, we primarily avoid any specialized prompting techniques (such as self-consistency Wang et al. (2022) and other related techniques). However, whenever we observe poor performance consistently across a particular model or setting, we employ specialized prompting techniques such as template pattern matching White et al. (2023) and chain-of-thought prompting Wei et al. (2022). Wherever applicable, we mention what prompting technique was used.

**Automated evaluation:** To create a fully automated pipeline, we also choose to evaluate the responses of the model against the actual ground-truth solution using OpenAI's GPT-3.5 model using a standardized prompt. This approach has various advantages. It allows us to avoid excessive manual effort for a negligible accuracy tradeoff. Secondly, it is capable of processing various output formats from various models equally well.

We employ this model to mark each LLM response as correct or wrong. However, to ensure that this evaluator itself does not induce loss of accuracy, we manually evaluate a balanced randomized LLM response set containing 150 responses and their corresponding ground truths, evenly distributed across all models and k-shot settings. We find that the evaluator can evaluate LLM responses with a confidence of 98%. In other words, in our paper, all LLM performances are guaranteed within 2% of error. The evaluator prompting format as well as manual evaluation for evaluator performance is included in the Appendix.

**Partial Credit evaluation:** Apart from an automated evaluation, we go one step further and manually evaluate problem categories 1 and 2 (except 2.4, where partial credit does not apply) for partial credit. We define partial credit as the fraction of nodes that the evaluator got correct out of all the nodes in the ground truth solution before predicting to a wrong node. This is only applicable in cases where the model does not get a full score of 1.0 from the primary evaluator. This metric allows us to evaluate the reasoning ability and problem solving technique of LLMs with greater granularity.

## 2.3 SELECTION OF LLMS

In order to include a broad range of models, we select five models from four families of models (in terms of architecture and training data/method). This includes – OpenAI's **GPT-3.5** and **GPT-4** OpenAI (2023), Anthropic's **Claude-2** Wu et al. (2023), Google's **Palm-2** Anil et al. (2023), and Meta's **Llama-2** Touvron et al. (2023) with 13B parameters. Since LLama-2 is a raw foundational model, we use an instruction-fine tuned version (Hermes) released by Nous Research.

## 3 RESULTS AND DISCUSSIONS

Using the prompting and evaluation technique defined above, we evaluate the various models over the various problems. We display results of graph reasoning for problem 1, problem 2 and problem 3 in tables 1, 2 and 3 respectively. The values represent averaged and normalized accuracies across 10 examples, and go from 0 to 1 in increments of 0.1. A model response is only marked correct, if it predicted the shortest/least-cost traversal path correctly and completely.

Table 1: Evaluation of models on problem category 1: tree-based graph traversals. The values are represented as A/B, where A depicts the average normalized binary accuracy (ranging from 0 to 1) for 10 examples per setting, and B depicts the average normalized value for partial credit for 10 examples per setting. * = a larger context variant of the corresponding LLM was used. △ = the prompt was too large to fit the context window of the model, hence the setting was omitted. ** = the model did not return any solution and refused to answer the question.

| | O(10) | | | O(20) | | | O(20) jumbled | | |
| Problem | 0-shot | 1-shot | 3-shot | 0-shot | 1-shot | 3-shot | 0-shot | 1-shot | 3-shot |
| --- | --- | --- | --- | --- | --- | --- | --- | --- | --- |
| | | | | | GPT-3.5 | | | | |
| 1.1 | 1.0 | 0.6/0.80 | 1.0 | 1.0 | 1.0 | 1.0* | 0.0/0.09 | 0.1/0.08 | 0.0/0.07* |
| 1.2 | 0.2/0.41 | 0.4/0.59 | 0.6/0.62 | 0.0/0.39 | 0.1/0.38 | 0.2/0.31* | 0.0/0.22 | 0.2/0.29 | 0.0/0.29* |
| 1.3 | 0.3/0.53 | 0.0/0.24 | 0.4/0.48 | 0.2/0.36 | 0.1/0.17 | 0.0/0.26* | 0.0/0.28 | 0.0/0.21 | 0.0/0.19* |
| 1.4 | 0.3/0.47 | 0.2/0.45 | 0.0/0.48 | 0.1/0.37 | 0.0/0.23 | 0.0/0.33* | 0.1/0.24 | 0.0/0.13 | 0.0/29* |
| | | | | | GPT-4 | | | | |
| 1.1 | 1.0 | 1.0 | 1.0 | 1.0 | 1.0 | 1.0 | 0.7/0.84 | 0.9/0.98 | 0.7/0.87 |
| 1.2 | 1.0 | 1.0 | 1.0 | 0.5/0.67 | 0.8/0.85 | 0.9/0.93 | 0.1/0.31 | 0.2/0.47 | 0.4/0.55 |
| 1.3 | 0.8/0.93 | 0.6/0.69 | 0.9/0.91 | 0.4/0.64 | 0.5/0.70 | 0.6/0.71 | 0.2/0.45 | 0.2/0.40 | 0.3/0.51 |
| 1.4 | 0.4/0.62 | 0.0/0.31 | 0.4/0.59 | 0.2/0.40 | 0.1/0.35 | 0.0/0.27 | 0.1/0.26 | 0.2/0.39 | 0.3/0.51 |
| | | | | | Claude-2 | | | | |
| 1.1 | 1.0 | 0.0/0.60 | 1.0 | 0.2/0.30 | 0.2/0.43 | 1.0 | 0.0/0.07 | 0.0/0.06 | 0.0/0.12 |
| 1.2 | 0.7/0.84 | 0.0/0.30 | 0.3/0.50 | 0.3/0.53 | 0.1/0.29 | 0.2/0.31 | 0.1/0.31 | 0.0/0.18 | 0.0/0.22 |
| 1.3 | 0.4/0.62 | 0.5/0.70 | 0.2/0.56 | 0.0/0.37 | 0.2/0.39 | 0.1/0.35 | 0.3/0.36 | 0.0/0.26 | 0.0/0.14 |
| 1.4 | 0.4/0.56 | 0.2/0.44 | 0.1/0.34 | 0.2/0.41 | 0.0/0.25 | 0.1/0.34 | 0.1/0.30 | 0.0/0.19 | 0.0/0.17 |
| | | | | | Llama-2 | | | | |
| 1.1 | 1.0 | 0.0/0.10 | 1.0 | 0.0/0.05 | 0.0/1.0 | △ | 0.0/0.06 | 0.0/0.05 | △ |
| 1.2 | 0.2/0.47 | 0.1/0.31 | 0.0/0.18 | 0.0/0.15 | 0.0/0.14 | △ | 0.0/0.11 | 0.0/0.16 | △ |
| 1.3 | 0.0/0.30 | 0.1/0.35 | 0.0/0.25 | 0.0/0.17 | 0.0/0.13 | △ | 0.0/0.14 | 0.0/0.16 | △ |
| 1.4 | 0.2/0.40 | 0.1/0.37 | 0.0/0.32 | 0.0/0.16 | 0.0/0.25 | △ | 0.0/0.15 | 0.0/0.17 | △ |
| | | | | | Palm-2 | | | | |
| 1.1 | 1.0 | 1.0 | 1.0 | 1.0 | ** | ** | 0.0/0.05 | 0.0/0.05 | 0.0/0.06 |
| 1.2 | 0.2/0.27 | 0.0/0.32 | 0.1/0.44 | 0.0/0.27 | ** | ** | 0.0/0.19 | 0.0/0.17 | ** |
| 1.3 | 0.1/0.52 | 0.2/0.53 | 0.2/0.50 | ** | ** | ** | ** | ** | ** |
| 1.4 | 0.0/0.35 | 0.1/0.38 | 0.0/0.34 | 0.0/0.10 | ** | ** | 0.0/0.08 | ** | ** |

### 3.1 GENERAL OBSERVATIONS

Before analysing the performance of various LLMs on different LLMs, we observe some overall characteristics of models. Particularly, we observe issues with Llama-2 and Palm-2. In the case of LLama-2, 3-shot setting for O(20) graphs could not be evaluated, because of the limited context-window length of the model (4,096 tokens). While this issue was also seen with the default GPT-3.5 model, a GPT-3.5 variant with a larger context window (16,383 tokens) was able to solve the issue.

In the case of Palm-2, we observe that beyond a certain length of input (in the case of 1-shot and 3-shot prompts), the model tends to not respond with any solution, but with a generic statement related to the model's inability to solve the problem.

### 3.2 OVERALL OBSERVATIONS FROM PROBLEM 1 AND PROBLEM 2

Through a first, glance, we can easily point out that all models generally perform better on tree-based graphs (problem 1) than grid-based graphs (problem 2). Further we notice a general drop in performance in random tree traversal (problem 1.2) across O(10) and O(20) graphs. **This clearly indicates that a greater number of average degrees of freedom for traversal per node has an inverse correlation with LLM reasoning capability.**

Table 2: Evaluation of models on problem category 2: grid-based graph traversals. The values are represented as A/B, where A depicts the average normalized binary accuracy (ranging from 0 to 1) for 10 examples per setting, and B depicts the average normalized value for partial credit for 10 examples per setting. * = a larger context variant of the corresponding LLM was used. △ = the prompt was too large to fit the context window of the model, hence the setting was ommitted. ** = the model did not return any solution and refused to answer the question.

| | O(10) | | | O(20) | | | O(20) jumbled | | |
|---|---|---|---|---|---|---|---|---|---|
| Problem | 0-shot | 1-shot | 3-shot | 0-shot | 1-shot | 3-shot | 0-shot | 1-shot | 3-shot |
| | | | | | GPT-3.5 | | | | |
| 2.1 | 0.1/0.33 | 0.2/0.46 | 0.3/0.41 | 0.1/0.21 | 0.1/0.28 | 0.1/0.28* | 0.1/0.14 | 0.0/0.13 | 0.0/0.12* |
| 2.2 | 0.0/0.40 | 0.0/0.19 | 0.0/0.37 | 0.0/0.30 | 0.0/0.12 | 0.1/0.32* | 0.0/0.13 | 0.1/0.13 | 0.0/0.13* |
| 2.3 | 0.1/0.40 | 0.10/0.36 | 0.0/0.37 | 0.0/0.14 | 0.0/0.17 | 0.0/0.22* | 0.0/0.13 | 0.0/0.09 | 0.0/0.11* |
| 2.4 | 0.0 | 0.0 | 0.0 | 0.0 | 0.0 | 0.3* | 0.0 | 0.0 | 0.0* |
| | | | | | GPT-4 | | | | |
| 2.1 | 0.3/0.46 | 0.3/0.48 | 0.4/0.51 | 0.1/0.36 | 0.1/0.29 | 0.0/0.21 | 0.0/0.14 | 0.0/0.23 | 0.0/0.16 |
| 2.2 | 0.0/0.44 | 0.4/0.56 | 0.2/0.37 | 0.1/0.42 | 0.1/0.31 | 0.3/0.54 | 0.0/0.14 | 0.0/0.26 | 0.0/0.23 |
| 2.3 | 0.2/0.52 | 0.0/0.36 | 0.2/0.40 | 0.1/0.34 | 0.2/0.49 | 0.1/0.28 | 0.0/0.16 | 0.0/0.16 | 0.0/0.17 |
| 2.4 | 0.3 | 0.3 | 0.4 | 0.4 | 0.4 | 0.4 | 0.4 | 0.9 | 0.6 |
| | | | | | Claude-2 | | | | |
| 2.1 | 0.0/0.40 | 0.3/0.44 | 0.0/0.36 | 0.0/0.29 | 0.1/0.20 | 0.0/0.22 | 0.0/0.16 | 0.0/0.13 | 0.1/0.16 |
| 2.2 | 0.1/0.34 | 0.2/0.41 | 0.1/0.31 | 0.0/0.19 | 0.3/0.23 | 0.0/0.21 | 0.0/0.17 | 0.1/0.12 | 0.0/0.13 |
| 2.3 | 0.1/0.33 | 0.1/0.29 | 0.0/0.24 | 0.0/0.17 | 0.0/0.18 | 0.0/0.26 | 0.0/0.12 | 0.0/0.13 | 0.0/0.12 |
| 2.4 | 0.0 | 0.1 | 0.3 | 0.0 | 0.1 | 0.3 | 0.0 | 0.3 | 0.4 |
| | | | | | Llama-2 | | | | |
| 2.1 | 0.0/0.16 | 0.2/0.44 | 0.1.0/0.24 | 0.0/0.18 | 0.1/0.12 | △ | 0.0/0.05 | 0.0/0.05 | △ |
| 2.2 | 0.0/0.22 | 0.0/0.40 | 0.1/0.52 | 0.0/0.18 | 0.0/0.36 | △ | 0.0/0.04 | 0.0/0.06 | △ |
| 2.3 | 0.0/0.14 | 0.0/0.22 | 0.0/0.12 | 0.0/0.08 | 0.0/0.12 | △ | 0.0/0.014 | 0.0/0.06 | △ |
| 2.4 | 0.1 | 0.8 | 0.0 | 0.0 | 0.0 | △ | 0.0 | 0.0 | △ |
| | | | | | Palm-2 | | | | |
| 2.1 | 0.0/0.22 | 0.5/0.64 | 0.2/0.42 | 0.0/0.06 | ** | ** | 0.0/0.04 | ** | ** |
| 2.2 | 0.0/0.21 | 0.0/0.24 | 0.0/0.30 | 0.0/0.04 | ** | ** | ** | ** | ** |
| 2.3 | 0.0/0.18 | 0.0/0.30 | 0.0/0.22 | ** | ** | ** | 0.0/0.08 | ** | ** |
| 2.4 | 0.0 | 0.0 | 0.0 | ** | ** | ** | ** | ** | ** |

Table 3: Evaluation of models on problem category 3: special problems. The values depict the average normalized value (ranging from 0 to 1) for 10 examples per setting. - = the setting is not applicable for a particular level. △ = the prompt was too large to fit the context window of the model, hence the setting was ommitted. ** = the model did not return any solution and refused to answer the question. Note that for problem 3.1, the random baseline is 0.5, since the problem solution is binary (True/False).

| | O(10) | | | O(20) | | | O(20) jumbled | | |
|---|---|---|---|---|---|---|---|---|---|
| Problem | 0-shot | 1-shot | 3-shot | 0-shot | 1-shot | 3-shot | 0-shot | 1-shot | 3-shot |
| | | | | | GPT-3.5 | | | | |
| 3.1 | 0.3 | 0.1 | 0.5 | - | - | - | - | - | - |
| 3.2 | 0.0 | 0.0 | 0.2 | 0.0 | 0.0 | 0.0 | 0.0 | 0.0 | 0.0 |
| | | | | | GPT-4 | | | | |
| 3.1 | 0.5 | 0.5 | 0.2 | - | - | - | - | - | - |
| 3.2 | 0.2 | 0.3 | 0.2 | 0.0 | 0.0 | 0.1 | 0.0 | 0.1 | 0.0 |
| | | | | | Claude-2 | | | | |
| 3.1 | 0.6 | 0.4 | 0.7 | - | - | - | - | - | - |
| 3.2 | 0.1 | 0.1 | 0.2 | 0.1 | 0.0 | 0.0 | 0.0 | 0.0 | 0.0 |
| | | | | | Llama-2 | | | | |
| 3.1 | 0.6 | 0.8 | 0.7 | - | - | - | - | - | - |
| 3.2 | 0.0 | 0.0 | 0.0 | 0.0 | 0.0 | △ | 0.0 | 0.0 | △ |
| | | | | | Palm-2 | | | | |
| 3.1 | 0.5 | 0.1 | 0.2 | - | - | - | - | - | - |
| 3.2 | 0.0 | 0.0 | 0.0 | 0.0 | ** | ** | 0.0 | ** | ** |

**Model specific observations:** We start by analyzing the reasoning capabilities of models on the most trivial task (problem 1.1). Except Llama-2 and Claude-2, all models can navigate through

linear graph traversal when the nodes are unjumbled. While Llama-2 consistently performs poorly for other problems as well, Claude-2's sub-standard performance on a trivial linear graph traversal is surprising, since it does show decent performance in more complex problems. This indicates a training distribution bias in Claude-2 that avoids purely sequential node traversals. **However, apart from GPT-4, no model can solve linear graph traversal when the nodes are jumbled (O(20) jumbled), depicting a lower logical problem solving capabilities in the models.**

For more complex problems (1.3 and 1.4), we observe some reasoning capability in GPT-4 and Claude-2. GPT-3.5 also demonstrates some reasoning capabilities, but this quickly declines as the order of the graph increases.

As for grid-based traversals, simpler grid traversals (problem 2.1) can be seen in GPT4 and Palm-2. Palm-2 demonstrates some reasoning abilities only in few-shot settings for smaller prompts. **However, adding more constraints to grid-traversal problems results in a quick drop in accuracy across all models.**

### 3.3 Weighted vs unweighted graph traversal

Adding weights to edges of a graph in a least-cost traversal problem imposes a significant constraint, since it requires an additional memory component to track the sum of weights. Unsurprisingly, we observe a general trend across all models of dropping performance as soon as weights are added to trees (problems 1.3 vs 1.4) and well as grids (problems 2.1 vs 2.2). **However, GPT-4 and Claude-2 demonstrate some ability to solve weighted traversal in few-shot settings, demonstrating their innate ability to numerical variable states in parallel to graph paths.** A quick chain-of-thought prompting Wei et al. (2022) in GPT-3.5 and Llama-2 demonstrates that the models tend to neglect weights and solve problems using an unweighted approach. In many of these cases, these models tend to respond with an algorithmic direction, but fails to provide any viable response, especially in 0-shot settings.

### 3.4 effect of k-shot prompting

Across all models and levels, we observe no generalized trend of performance improvement in few-shot prompting vs 0-shot prompting. Infact, we observe a slight drop in performance in 3-shot prompts compared to one-shot prompt in some cases. In general, while few-shot prompting is helpful in response format shaping, **it is clear that few-shot prompts does not contribute towards logical providing any reasoning context, and infact, may affect reasoning capability as a counter effect of template formatting.**

### 3.5 effect of jumbling node order

Poor performance over jumbled node sequences can be observed consistently across all models and levels. Even linear graph traversal, an extremely trivial traversal problem, proves to be unsolvable by all models. **This proves a strong bias within models towards the expectation of nodes to appear in some form of order, even in few-shot settings.** The only exception is GPT-4, which attempts to solve graphs through valid logical principles. However, this approach is again, majorly unsuccessful across grid-based traversals.

### 3.6 Positive response bias within LLMs

Through a special problem (problem 2.4), we observe that all models also consistently fail to recognize when no solution is possible. An interesting observation is that models fail to recognize when no valid solution is possible even in few-shot prompt settings, considering it is expected that this method should induce a formatting bias in the response. **This indicates a clear training regime flaw among models to prefer avoiding empty responses at the cost of predicting wrong solutions**. However, given that GPT-4 and Claude-2 do possess some valid logical reasoning and state tracking abilities, we observe some ability to predict the inexistence of any valid solution in these models.

### 3.7 Performance of LLMs over special problems

As mentioned earlier, in this paper, the eulerian problem (problem 3.1) is a binary problem, and thus a random classifier is expected to perform with an accuracy of 0.5. We observe slightly better performance than random only in the case of Claude-2 and Llama-2. **However, in all other models, the performance over identification of a valid eulerian graph is equal to or less than random.** . Analysis of the responses manually highlights that models tend to respond with a positive response.

### 3.8 Analysis of partial credit across models

Binary accuracy is a highly strict metric, since a single wrong node can mark the entire response wrong. Keeping this in mind, we analyse the responses from models using partial credit as well. Particularly, we are interested in analysing interesting trends that were not highlighted by the accuracy metric.

Carefully observing the results for graphs with jumbled node orders, we observe that GPT-3.5, GPT-4 as well as Claude-2, start in the right direction, but tend to deviate after 4-8 nodes in the case of $O(20)$ tree based graphs. This number further diminishes to 2-3 nodes in the case of $O(20)$ grid based graphs. **Thus, even the most advanced models like GPT-3.5, GPT-4, and Claude-2 can be estimated to have a memory depth of less than 10 variable states.** The size of the model (GPT-4 > Claude-2 > GPT-3.5) has a direct correlation with the number of memory states that it can maintain simultaneously. This is consistent with the comparison of partial credit across both $O(10)$ graphs and $O(20)$ graphs.

Secondly, we also observe consistently that the partial credit for 1-shot setting for all models is higher than 3-shot setting. Hence, it indicates that multiple examples in the prompt result in "overfitting" of the model. Example responses from the few-shot prompts push LLMs to narrate responses from its training data, instead of logical calculations.

## 4 Summary

Through our

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
