# OpenReview forum: "Exploring the Limitations of Graph-based Logical Reasoning in Large Language Models"
_ICLR.cc/2024/Conference — ICLR 2024 Conference Withdrawn Submission_

### Official Review · Reviewer_WkiA · 2023-10-31

**Soundness:** 2 fair
**Presentation:** 1 poor
**Contribution:** 1 poor
**Rating:** 3
**Confidence:** 4

**Summary:**

The paper reviews the power of widespread LLMs with respect to several graph-traversal or related problems. The authors propose a benchmark of 10 problems, for which several graphs are created. Each graph in each problem is submitted to LLMs, and authors provide details. on the resulting accuracy, compared to the ground truth (minus some error because authors use LLM themselves to compare the response of models against the ground truth). The result point out to limited power of LLMs, and authors show that branching truly creates complications on the way LLMs process the results.

**Strengths:**

* Good standardized benchmark provides unbiased evaluation of each LLM.
* Interesting insights relating the degrees of freedom of each problem.

**Weaknesses:**

* The scope of the paper is somewhat limited, and these results are expected to what one understands of LLMs through general knowledge. This feels more like a workshop paper with respect to the scope and impact.
* The benchmark always assume graphs are given as adjacency matrices. Yet, in real life, most graphs are not stored in this way, but rather as adjacency lists (or, equivalently, storing adjacency matrices as sparse matrices).
* Authors claim "... that a greater number of average degrees of freedom for traversal per node has an inverse correlation with LLM reasoning capability". but they get this out of comparing an experiment with 10 nodes and an experiment of 20 nodes. This is not enough to conclude the aforementioned claim, as there could be several other explanations for the decreased performance, such as more complexity out of trying to decode the matrix with more nodes, or failure to work with more total memory. At least, to verify this claim, authors should maintain total number of nodes but increase the level of branching encountered in the traversal (for example in problem 1.2).

**Questions:**

I would be interested in previous motivation for this paper in the literature, or some previous evidence that the impact of these results merit publication in ICLR.

---

### Official Review · Reviewer_eQSj · 2023-11-01

**Soundness:** 2 fair
**Presentation:** 2 fair
**Contribution:** 2 fair
**Rating:** 3
**Confidence:** 4

**Summary:**

This paper evaluates the logical reasoning depth of five LLMs (GPT-4, GPT-3.5, Claude-2, Llama-2, and Palm-2) for graph traversal problems. It presents 10 complex graph problems and assesses the models' performance. Results show that most LLMs lack strong reasoning abilities, with performance declining as graph complexity increases. The use of more k-shot prompts also negatively impacts reasoning.

**Strengths:**

This paper evaluates the complex reasoning ability of LLMs from the perspective of graph reasoning, which is rational.

**Weaknesses:**

1. This empirical paper focuses on evaluating the capabilities of LLMs. However, the analyzed properties of LLMs are unsurprising and do not provide new insights to the research community.
2. The paper suggests a negative impact on reasoning abilities with an increase in the number of k-shot examples. However, the authors only tested experiments with 1-2-3 shots, and more experiments are needed to support this conclusion and provide explanations.
3. The summary section of the paper remains incomplete and the paper requires further improvement before publication.

**Questions:**

Please refer to weakness point 2.

---

### Official Review · Reviewer_aBCN · 2023-11-01

**Soundness:** 2 fair
**Presentation:** 1 poor
**Contribution:** 1 poor
**Rating:** 1
**Confidence:** 4

**Summary:**

This paper proposes a method that uses graphs to evaluate the reasoning ability of large language model. The authors evaluate 5 different LLMs and get some observations.

**Strengths:**

* The topic of reasoning ability of LLMs is interesting.

**Weaknesses:**

* The bad written and missed material. The summary of the article is not written. In section 2.2, the authors mentioned that the format of prompts is included in the Appendix, but they fail to submit any supplemental material.
* Limited novelty. There are several works and benchmarks which can be used to evaluate the reasoning ability and other ability of LLMs, such as PIQA[1], ARC[2] and Plan[3]. And the authors should have a section to introduce these related works and analysis the differences between the proposed method and these existing papers.
* There are some strange definitions which counter to the common ones. In figure1. the example graph in 1.3. and 1.4. represent some tree-based traversal problem but tree is exactly a kind of acyclic graph.
* The difference of proposed problem1 and problem2 is not obvious. Because both the graph of problem1 and problem2, the representation in a computer is similar, which consists of a adjacency matrix and an edge weights list.
* The section 3.4 mentioned that it is clear that few-shot prompts does not contribute towards logical providing any reasoning context. My main concern is that there seems to be no clear trend proving this from the experimental results. For example, in table 1, there are 36% of results that 3-shot get the best score and 50% regardless the special cases such as out of the context window.



[1] Bisk Y, Zellers R, Gao J, et al. Piqa: Reasoning about physical commonsense in natural language[C]//Proceedings of the AAAI conference on artificial intelligence. 2020, 34(05): 7432-7439.

[2] Clark P, Cowhey I, Etzioni O, et al. Think you have solved question answering? try arc, the ai2 reasoning challenge[J]. arXiv preprint arXiv:1803.05457, 2018.

[3] Valmeekam K, Olmo A, Sreedharan S, et al. Large Language Models Still Can't Plan (A Benchmark for LLMs on Planning and Reasoning about Change)[J]. arXiv preprint arXiv:2206.10498, 2022.

**Questions:**

see weaknesses

---

### Official Review · Reviewer_HHfy · 2023-11-04

**Soundness:** 3 good
**Presentation:** 2 fair
**Contribution:** 2 fair
**Rating:** 3
**Confidence:** 4

**Summary:**

The authors examine the sophistication of path discovery capabilities for 5 different LLMs (GPT-4, GPT-3.5, Claude-2, Llama-2 and Palm-2), a computational foundation of graph reasoning. Ten (10) distinct problems in graph traversal are examined using synthetically generated graphs; each problem represents an increasing levels of complexity in graph reasoning.

**Strengths:**

Originality: A competed and well documented experimental study of this type would be a contribution to the community understanding of the ability of large language models.

Quality: the study design seems sound, although experiments are not documented sufficiently for evaluation.  Some experiments seem incomplete.  The paper is incomplete, with some sections missing.

Clarity: The authors describe their intent clearly.

Significance: This experiment would provide an interesting comparison point on LLM as graph-solver.

**Weaknesses:**

The work is incomplete.

The literature review may be incomplete, depending on the final conclusions of the paper.

The Quality, Clairty, and Significance of a final paper are hard to judge in its current state.

**Questions:**

1. Do these experiments uncover interesting behavior of transformers in general, or just the models studied?

I can envision a completed version of this paper that would result in a higher rating, but in its current incomplete form I'm going to recommend rejection.